SPECIAL ISSUE
CILIA AND FLAGELLA: FROM BASIC BIOLOGY TO DISEASE

# Phosphorylation-induced SUMOylation promotes Ulk4 condensation at the ciliary tip to transduce Hedgehog signal

Mengmeng Zhou[1], Yuhong Han[1] and Jin Jiang[1,2,*]

## ABSTRACT

Hedgehog (Hh) signaling controls embryonic development and adult tissue homeostasis through the Gli family of transcription factors. In vertebrates, Hh signal transduction depends on the primary cilium, where Gli proteins are thought to be activated at the ciliary tip, but the underlying mechanism has remained poorly understood. Here, we provide evidence that two Unc-51-like kinase (Ulk) family members, Stk36 and Ulk4, regulate Gli2 ciliary tip localization and activation through phosphorylation and SUMOylation-mediated condensation in response to the Hh family protein Shh. We find that Stk36-mediated phosphorylation of Ulk4 promotes its SUMOylation in response to Shh, and the subsequent interaction between SUMO and a SUMO-interacting-motif (SIM) in the C-terminal region of Ulk4 drives Ulk4 self-assembly to form biomolecular condensates that also recruit Stk36 and Gli2. SUMOylation or SIM-deficient Ulk4 failed to accumulate at ciliary tip to activate Gli2 whereas phospho-mimetic mutation of Ulk4 sufficed to drive Ulk4, Stk36 and Gli2 condensation at ciliary tip, leading to constitutive Shh pathway activation in a manner dependent on Ulk4 SUMOylation. Taken together, our results suggest that phosphorylation-dependent SUMOylation of Ulk4 promotes kinase–substrate condensation at ciliary tip to transduce the Hh signal.

KEY WORDS: Hedgehog, Gli, Ulk4, Phosphorylation, SUMOylation, Primary cilium

## INTRODUCTION

The Hh family of glycoproteins plays essential roles in embryonic development and adult tissue homeostasis (Briscoe and Therond, 2013; Jiang and Hui, 2008). Aberrant Hh signaling contributes to a wide range of human diseases including birth defects and cancer (Jiang, 2022; Nieuwenhuis and Hui, 2005; Villavicencio et al., 2000). Hh signaling in vertebrates requires primary cilia (Goetz and Anderson, 2010). In response to Hh stimulation, key Hh pathway components, including the GPCR family protein and Hh signal transducer Smoothened (Smo), as well as the pathway transcriptional factors Gli proteins, accumulate in primary cilia (Chen et al., 2009, 2011; Corbit et al., 2005; Haycraft et al., 2005;

Rohatgi et al., 2007; Tukachinsky et al., 2010). Gli proteins are thought to be activated at the tip of primary cilia because Gli2 variants that fail to accumulate in the cilia cannot be activated both in cultured cells and in mouse embryos (Han et al., 2017; Liu et al., 2015; Santos and Reiter, 2014); however, the underlying mechanisms that control ciliary tip localization and activation of Gli proteins have remained poorly understood.

Our previous studies have shown that two Ulk family kinases, Ulk3 and Stk36, act in parallel to convert full-length Gli2 (Gli2$^F$) into its activator form (Gli2$^A$) by directly phosphorylating Gli2 at multiple sites in mammalian cells (Han et al., 2019; Zhou et al., 2022; Zhou and Jiang, 2022). Another Ulk family member Ulk4, a pseudokinase, acts as a genetic modifier of the holoprosencephaly phenotype caused by impaired Shh pathway activity and physically interacts with Stk36 (Mecklenburg et al., 2021; Preuss et al., 2020). Our recent study demonstrated that Ulk4 acts in conjunction with Stk36 to promote Gli2 phosphorylation and Hh pathway activation by forming a complex with Stk36 and regulating Stk36 ciliary accumulation (Zhou et al., 2023). Interestingly, we found that Stk36 and Ulk4 depend on each other for their ciliary localization and that ciliary tip localization of Ulk4 depends on Stk36-mediated phosphorylation of Ulk4 on T1021 and T1023 (T1021/T1023) (Zhou et al., 2023).

Here, we provide evidence that phosphorylation of Ulk4 by Stk36 drives its phase separation to form biomolecular condensates that recruit Stk36, Gli2 and Sufu. We find that Stk36-mediated phosphorylation of Ulk4 at T1021/T1023 promotes its SUMOylation on an adjacent site (K1030) in response to Shh, and that interaction between SUMO and a SUMO-interacting motif (SIM) in the C-terminal region of Ulk4 drives Ulk4 self-association to form biomolecular condensates. SUMOylation or SIM-deficient Ulk4 fails to accumulate at ciliary tip and is thus unable to activate Gli2. Remarkably, a phospho-mimetic Ulk4 variant can form condensates that recruit Stk36, Gli2 and Sufu at ciliary tip in the absence of Shh, leading to constitutive Gli2 phosphorylation and pathway activation in a manner dependent on Ulk4 SUMOylation.

## RESULTS

### Phosphorylation of Ulk4 is sufficient to drive its ciliary tip accumulation

Our previous study has revealed that phosphorylation of Ulk4 by Stk36 is necessary for its ciliary tip accumulation (Zhou et al., 2023). To determine whether Ulk4 phosphorylation is sufficient to drive its ciliary tip accumulation, we generated a phospho-mimetic form of human Ulk4 by replacing two previously identified Stk36 sites (T1021 and T1023) with glutamic acid residues (Ulk4-EE). C-terminally HA-tagged wild type Ulk4 (Ulk4-WT) and Ulk4-EE were expressed in NIH3T3 cells via lentiviral infection, followed by Shh stimulation. In this experiment, endogenous Ulk4 was depleted by shRNA targeting the mouse Ulk4 to avoid its interreference with the ciliary localization of exogenously expressed Ulk4 in response

[1]Department of Molecular Biology, University of Texas Southwestern Medical Center, Dallas, TX 75390, USA. [2]Department of Pharmacology, University of Texas Southwestern Medical Center, Dallas, TX 75390, USA.

*Author for correspondence ( jin.jiang@utsouthwestern.edu)

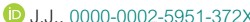 J.J., 0000-0002-5951-372X

Journal of Cell Science

to Shh. Of note, human Ulk4 was used to generate all the Ulk4 expression constructs used for this study because it is resistant to the shRNA targeting the mouse Ulk4. To avoid overexpression, we used the lentiviral vector carrying a weak promoter to express tagged Ulk4 at low levels, so that they are still regulated by Shh. Western blot analysis revealed that the expression levels of exogenously expressed Ulk4-WT and Ulk4-EE were comparable to that of endogenous protein (Fig. S1). As shown in Fig. 1A, Ulk4-WT–HA accumulated at ciliary tip only after Shh stimulation whereas Ulk4-EE–HA accumulated at ciliary tip regardless of Shh treatment. Quantification of ciliary tip Ulk4 signals revealed that Shh dramatically increased Ulk4-WT–HA levels but had little, if any, effect on Ulk4-EE–HA levels at ciliary tips (Fig. 1B), suggesting that Stk36-mediated phosphorylation of Ulk4 is sufficient to promote its ciliary tip accumulation.

We next asked whether Ulk4-EE could bring Stk36 to the ciliary tip in the absence of Shh. NIH3T3 cells were infected with lentivirus expressing Myc-tagged Stk36 (Myc–Stk36) and Ulk4-WT–HA or Ulk4-EE–HA, followed by immunostaining with antibodies against the epitope tags to monitor ciliary accumulation of Myc-Stk36 and Ulk4-WT–HA or Ulk4-EE–HA. Cells co-expressing Myc–Stk36 and Ulk4-WT–HA showed no ciliary tip accumulation of either Ulk4-WT–HA or Myc–Stk36 (Fig. 1C,D). By contrast, cells co-expressing Myc–Stk36 and Ulk4-EE–HA showed ciliary tip accumulation of both Ulk4-EE–HA and Myc–Stk36 (Fig. 1C,D), suggesting that Ulk4-EE could bring Stk36 to the ciliary tip even in the absence of Shh.

### Gli2 ciliary tip accumulation is promoted by Stk36 and Ulk4

In response to Shh, Gli proteins and their binding partner and Hh pathway inhibitor Sufu accumulate at ciliary tip while Smo

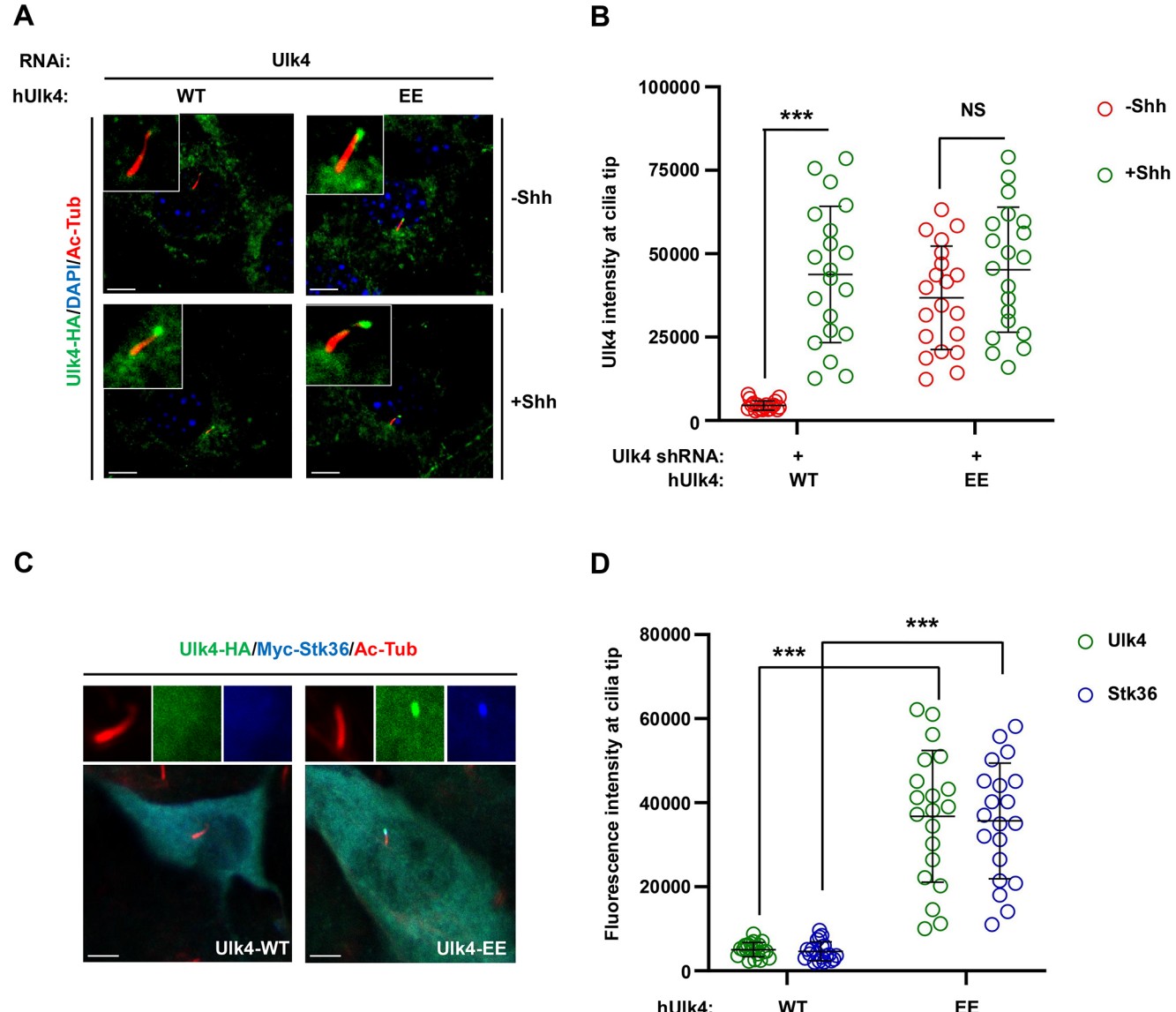

**Fig. 1. Stk36-mediated phosphorylation of Ulk4 suffices to drive its ciliary tip accumulation.** (A,B) Representative images of immunostaining (A) and quantification (B) of ciliary tip localized human Ulk4–HA (green in A) in NIH3T3 cells infected with mouse Ulk4 shRNA and lentivirus expressing human Ulk4-WT–HA or Ulk4-EE–HA in the presence or absence of Shh. (C,D) Representative images of immunostaining (C) and quantification (D) of ciliary tip-localized Ulk4–HA (green in C) or Myc–Stk36 (blue in C) in NIH3T3 cells infected with Ulk4-WT–HA or Ulk4-EE–HA lentivirus. Primary cilia were marked by acetylated tubulin (Ac-tub) staining (red in A and C). The intensity of ciliary-localized Ulk4–HA or Myc–Stk36 was measured by ImageJ. 20 cells were randomly selected from each experimental group for quantification. Data are mean±s.d. ***P<0.001; NS, not significant (unpaired two-tailed Student's t-test). Results in B and D are representatives of three independent experiments. Scale bars: 5 μm.

accumulates along the cilium axoneme (Chen et al., 2009, 2011; Corbit et al., 2005; Haycraft et al., 2005; Rohatgi et al., 2007; Tukachinsky et al., 2010). The mechanisms that control ciliary tip localization of Gli proteins are poorly understood although Kif7–Gli interaction has been implicated as playing an important role (Haque et al., 2022). The observations that Stk36 and Ulk4 colocalized with Gli2 at ciliary tip and that Ulk family kinases, including Ulk3 and Stk36, acted upstream of Gli2 to promote Shh pathway activation let us to speculate that Gli ciliary tip localization could be regulated by these kinases. Indeed, as shown in Fig. 2A,B, Shh-induced ciliary accumulation of Gli2 and Sufu was diminished in Stk36 and Ulk3 double knockout (DKO) NIH3T3 cells (NIH3T3$^{DKO}$) (Han et al., 2019). Expression of wild-type Stk36 (Stk36-WT) but not its kinase inactive form (Stk36-AA) restored ciliary tip accumulation of both Gli2 and Sufu in response to Shh (Fig. 2A,B). Furthermore, expression of Ulk4-EE but not Ulk4-WT in NIH3T3$^{DKO}$ cells restored ciliary tip localization of both Gli2 and Sufu, which correlated with ciliary localization of Ulk4-EE (Fig. 2C–F). These results suggest that ciliary tip localization of Gli2 and Sufu is promoted by Stk36 and Ulk4.

## Phosphorylation drives Ulk4 condensation

The ciliary tip accumulation of Hh signaling components resembles the biomolecular condensate formation process, which depends on multivalent interactions and protein concentrations (Banani et al., 2017; Shin and Brangwynne, 2017). Indeed, Stk36, Ulk4, Gli and Sufu physically interact with one another (Han et al., 2015; Preuss et al., 2020; Zhou et al., 2023). Upon Hh stimulation, Smo is accumulated and activated in primary cilia, which might result in local activation of Stk36 kinase and Stk36-mediated phosphorylation of Ulk4, leading to the formation of Ulk4 and Stk36 condensates that recruit Gli2 and Sufu. To test this hypothesis, we asked whether Stk36-mediated phosphorylation of Ulk4 can drive Stk36 and Ulk4 condensation in the cytoplasm when both proteins were overexpressed in HEK293T cells. Constitutively active Stk36 (Stk36-EE; S151E/T154E) or its kinase inactive form (Stk36-AA; S151A/T154A) was transfected into HEK293T cells together with Ulk4-WT (Zhou et al., 2023). Co-expression of Stk36-EE with Ulk4-WT led to the formation of Ulk4 puncta that also contained Stk36 (Fig. 3A). However, co-expression of Stk36-AA with Ulk4-WT failed to induce the formation of Ulk4 and Stk36 puncta (Fig. 3A). The formation of Ulk4 and Stk36 puncta depends on phosphorylation of Ulk4 by Stk36 because HEK293T cells co-expressing a phospho-deficient form of Ulk4 (Ulk4-2TA; T1021A/T1023A) and Stk36-EE failed to form Ulk4 and Stk36 puncta (Fig. 3B). Strikingly, HEK293T cells expressing Ulk4-EE formed Ulk4 condensates that also recruited endogenous Gli2 and Sufu into the condensates whereas cells expressing Ulk4-WT failed to do so (Fig. 3C). These results suggest that phosphorylation of Ulk4 by Stk36 promotes the formation of Ulk4 condensates that can recruit its interaction proteins, including Stk36, Gli2 and Sufu.

## STK36 and Ulk4 forms gel-like or solid-state condensates

Most membraneless biomolecular condensates form through liquid–liquid phase separation (LLPS), which can be disrupted by 1,6-hexanediol (McKnight, 2024; Sabari et al., 2018). Indeed, condensates of GFP-tagged YAP (GFP–YAP; YAP is also known as YAP1), which is formed via LLPS (Cai et al., 2019), were disrupted by treating cells with 5% 1,6-hexanediol for 5 min (Fig. 4A). To determine the biomaterial property of the condensates organized by Ulk4, we treated HEK293T cells that contained GFP–Stk36 and Ulk4-EE condensates with 5% 1,6-hexanediol for

5–30 min. We found that GFP–Stk36 and Ulk4-EE condensates were resistant to 1,6-hexanediol even after treatment for 30 min (Fig. 4A; Fig. S2). We next carried out fluorescence recovery after photobleaching (FRAP) experiments. In contrast to GFP–YAP whose signals recovered 10 s after photobleaching, GFP–Stk36 signals exhibited little if any recovery 300 s after photobleaching (Fig. 4B,C). These observations suggest that Stk36 and Ulk4 condensates are more rigid than those formed by LLPS and might resemble gel-like or solid-state condensates (Putnam et al., 2019; Woodruff et al., 2017). In line with this finding, our recent study showed that Ulk3 as well as the *Drosophila* Stk36 homolog Fused (Fu) also form gel-like or solid-state condensates (Han et al., 2025).

## Shh induces SUMOylation of Ulk4 dependent on Stk36-mediated phosphorylation

We next asked how Ulk4 phosphorylation promotes its condensation. Phosphorylation can regulate other post-translational modifications (PTMs), such as SUMOylation, and SUMOylation has been implicated in the formation of biomolecular condensates (Keiten-Schmitz et al., 2021; Shen et al., 2006; Yang and Gregoire, 2006). Interestingly, like Ulk4-mediated condensation, condensates formed by a synthetic construct containing multiple SUMO and SIM domains, RFP–SUMO6-SIM10 (Banani et al., 2016), were also resistant to 1,6-hexanediol treatment (Fig. 4A; Fig. S2). Therefore, we asked whether phosphorylation induces Ulk4 condensation through promoting its SUMOylation. Co-expression of Ulk4–HA with Flag-tagged (Fg) Fg–Stk36-EE but not with Fg–Stk36-AA in HEK293T cells resulted in a mobility shift of Ulk4–HA indicative of SUMO conjugation (Fig. 5A, top panel). Probing with anti-SUMO1 antibody confirmed that higher molecular mass bands contained SUMOylated Ulk4 (Fig. 5A, bottom panel). Mutating the Stk36 sites on Ulk4 (Ulk4-2TA) abolished Ulk4 SUMOylation stimulated by Stk36-EE (Fig. 5B). By contrast, Ulk4-EE was SUMOylated without co-expression of Stk36-EE, and SUMOylation of Ulk4-EE was abolished by a SUMOylation inhibitor TAK-981 (Fig. 5C) (Langston et al., 2021). These results demonstrated that Stk36-mediated phosphorylation at T1021/T1023 can induce Ulk4 SUMOylation. Of note, SUMOylated Ulk4 tended to form high molecular mass species on the SDS-PAGE gels, a phenomenon also observed for SUMOylated nuclear hormone receptors (Poukka et al., 2000; Tian et al., 2002).

To determine whether Shh-induced SUMOylation of Ulk4 was dependent on its phosphorylation by Stk36, Ulk4-WT or Ulk4-2TA was expressed in NIH3T3 cells via lentiviral infection, followed by Shh stimulation. As shown in Fig. 5D, Shh induced SUMOylation of Ulk4-WT but not Ulk4-2TA.

SUMOylation sites have consensus motifs ψKxE/D or E/DxKψ, where ψ is a large hydrophobic amino acid and x is any amino acid (Matic et al., 2010). SUMOylation of many proteins is promoted by phosphorylation on sites adjacent to the SUMOylation sites (Yang and Gregoire, 2006). Inspection of Ulk4 C-terminal region identified a SUMOylation consensus site (ESK$_{1030}$L) adjacent to the Stk36 phosphorylation sites (Fig. 5E). To determine whether phosphorylation of Ulk4 at T1021/T1023 could promote SUMOylation at K1030, we carried out *in vitro* SUMOylation experiments. GST fusion proteins containing a Ulk4 fragment that encompassed both the Stk36 sites and the putative SUMOylation site (GST-Ulk4C; Fig. 5E) were incubated with immunopurified Stk36-EE and recombinant casein kinase 1 (CK1) for *in vitro* phosphorylation, followed by incubation with SUMO1 and SUMO enzymes (E1 and E2) for *in vitro* SUMOylation. As shown in Fig. 5F, GST–Ulk4C was SUMOylated after kinase

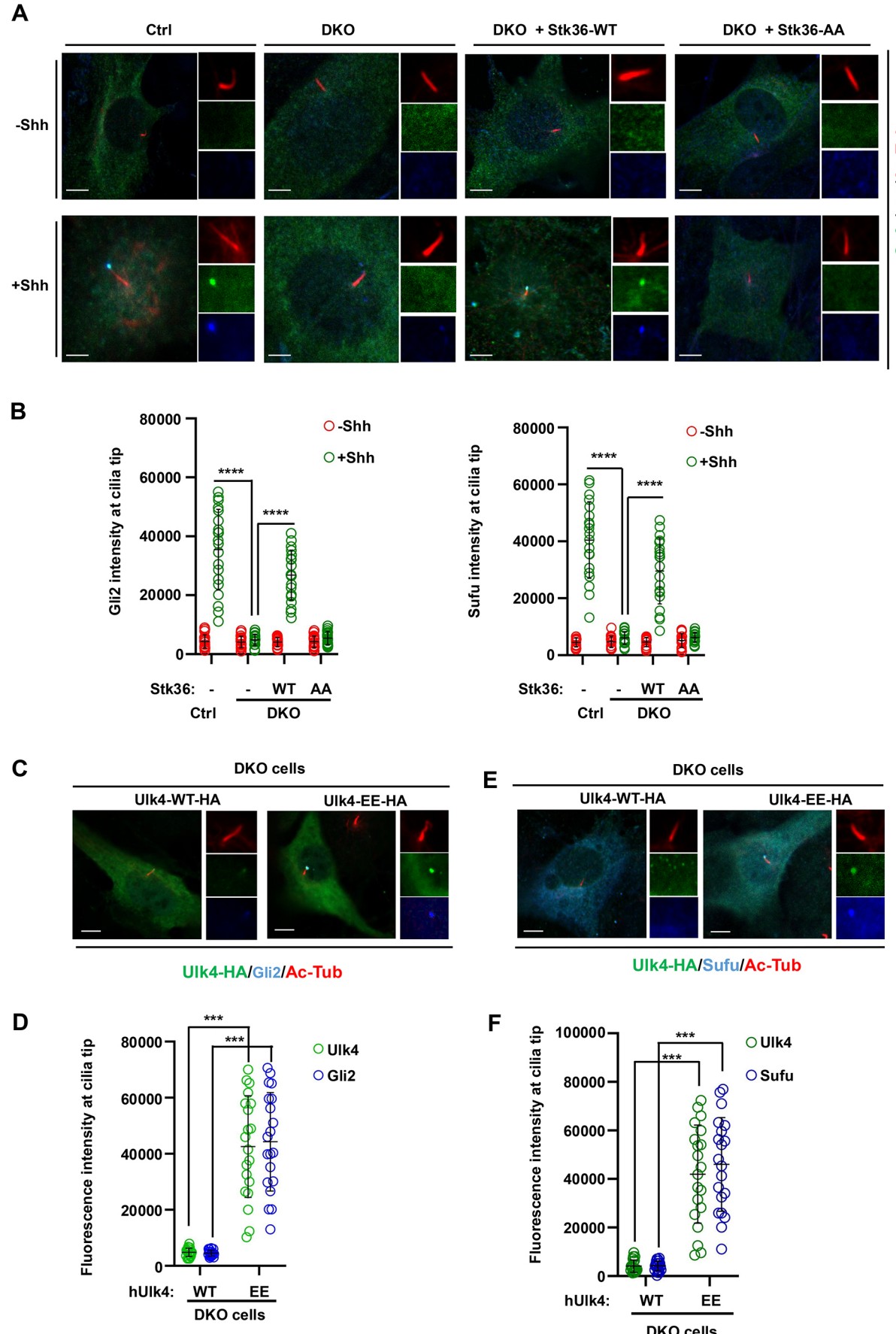

**Fig. 2.** See next page for legend.

**Fig. 2. Stk36 and Ulk4 promote ciliary tip accumulation of Gli2 and Sufu.** (A,B) Representative images of immunostaining (A) and quantification (B) of ciliary tip localized endogenous Sufu (green in A) or Gli2 (blue in A) in control or *Ulk3^{KO} Stk36^{KO}* (DKO) NIH3T3 cells infected with Stk36-WT or Stk36-AA lentivirus and treated with or without Shh-N. Data are mean±s.d. ****P<0.0001 (one-way ANOVA test with Dunnett's multiple comparisons test). Results in are representatives of three independent experiments. (C–F) Representative images of immunostaining (C,E) and quantification (D, F) of ciliary tip localized Ulk4-HA (green in C, E), Gli2 (blue in C) or Sufu (blue in E) in DKO cells infected with Ulk4-WT–HA or Ulk4-EE–HA lentivirus. Data are mean±s.d. ***P<0.001 (unpaired two-tailed Student's *t*-test). Results are representatives of three independent experiments. Primary cilia were marked by acetylated tubulin (Ac-tub) staining (red in A, C and E). The intensity of ciliary-localized Sufu, Gli2 or Ulk4–HA was measured by ImageJ. 20 cells were randomly selected from each experimental group for quantification. Scale bars: 5 µm.

phosphorylation. Of note, CK1 was included to boost Stk36 kinase activity in the *in vitro* phosphorylation assay because the active loop phosphorylation site T158 conforms to the CK1 consensus site and previous studies have revealed that phosphorylation of this conserved site in *Drosophila* Stk36 homolog Fused (Fu) is essential for Fu kinase activation (Shi et al., 2011; Zhou and Kalderon, 2011). Mutation of either K1030 to an arginine residue (KR) or T1021/T1023 to alanine residues (2TA) abolished *in vitro* SUMOylation of GST–Ulk4C (Fig. 5F). Furthermore, the K1030R mutation abolished Stk36-induced SUMOylation of Ulk4 and the constitutive SUMOylation of Ulk4-EE (Ulk4-EE-KR) in HEK293T cells, as well as Shh-induced Ulk4 SUMOylation in NIH3T3 cells (Fig. 5G–I). Taken together, these results demonstrate that Shh induces SUMOylation of Ulk4 on K1030 in a manner that is dependent on its phosphorylation at T1021/T1023.

## SUMOylation promotes Ulk4 self-association via SUMO–SIM interaction

SUMOylation mediates protein–protein interaction by binding to a SIM, which consists of a stretch of hydrophobic amino acids adjacent to acidic residues (Song et al., 2004). Interestingly, Ulk4 contains a SIM ($^{1145}$EDLLLL$^{1151}$DLE) located C-terminally to the phosphorylation-mediated SUMOylation site (Fig. 5E). To determine whether SUMOylation of Ulk4 promotes its self-association through interacting with this SIM, we examined the interaction between Flag-tagged Ulk4-EE (Ulk4-EE–Fg), which is constitutively SUMOylated, or Ulk4-EE-KR–Fg, which is no longer SUMOylated (Fig. 5H), with HA-tagged wild type (Ulk4-WT–HA) or SIM-deficient Ulk4 (Ulk4-ΔSIM–HA; Fig. 5E). These constructs were transfected into HEK293T cells, followed by co-immunoprecipitation (co-IP) and western blot analysis. As shown in Fig. 5J, Ulk4-EE–Fg formed a complex with Ulk4-WT–HA, which was diminished when SIM was mutated in Ulk4-WT–HA (Ulk4-ΔSIM–HA) or the SUMOylation site was mutated in Ulk4-EE-Fg (Ulk4-EE-KR–Fg). We also generated a C-terminal fragment of Ulk4-EE from amino acids (aa) 1004–1275 of (C272-EE–Fg). C272-EE–Fg formed a complex with Ulk4-WT–HA albeit at lower affinity compared with Ulk4-EE-Fg; however, C272-EE–Fg failed to form a complex with Ulk4-ΔSIM–HA (Fig. 5K). These observations suggest that (1) SUMO–SIM interaction facilitates Ulk4 self-association through its C-terminal region and (2) the N-terminal region of Ulk4 can also mediate Ulk4 self-association independent of SUMO–SIM interaction.

## SUMOylation and SIM are essential for Ulk4 condensation and Ulk4-mediated Shh pathway activation

To determine whether SUMOylation and SIM mediates Ulk4 condensation as is the case for promyelocytic leukemia (PLM)

bodies (Shen et al., 2006), we co-expressed Stk36-EE with wild-type (Ulk4-WT), SUMOylation deficient (Ulk4-K1030R) or SIM-mutated Ulk4 (Ulk4-ΔSIM) in HEK293T cells. As shown in Fig. 6A–C, whereas Ulk4-WT formed condensates that contained Stk36-EE and Gli2, neither Ulk4-K1030R nor Ulk4-ΔSIM formed condensates but rather exhibited diffused distribution in HEK293T cells.

Next, we expressed human Ulk4-WT, Ulk4-K1030R or Ulk4-ΔSIM via lentiviral infection in NIH3T3 cells with endogenous Ulk4 depleted by shRNA and treated the cells with or without Shh. Western blot analysis revealed that these Ulk4 constructs were expressed at levels comparable to the endogenous Ulk4 level (Fig. S3). In contrast to Ulk4-WT, which was accumulated at the ciliary tip after Shh treatment, neither Ulk4-K1030R nor Ulk4-ΔSIM accumulated at ciliary tip in response to Shh stimulation (Fig. 6D,E). Whereas Ulk4-WT rescued Gli2 phosphorylation, *Ptch1* and *Gli1* expression induced by Shh in Ulk4-depleted cells, neither Ulk4-K1030R nor Ulk4-ΔSIM rescued Gli2 phosphorylation and Shh target gene expression (Fig. 6F–H). These results suggest that Shh-induced Ulk4 SUMOylation and subsequent SUMO–SIM interaction is essential for Ulk4 condensation at ciliary tip and its role in mediating Shh pathway activation.

## Ulk4-EE exhibits constitutive pathway activity that is dependent on its SUMOylation

Ulk4-EE formed condensates in the cytoplasm when expressed in HEK293T cells and accumulated at the ciliary tips in NIH3T3 cells in the absence of Shh. Treating cells with the SUMOylation inhibitor TAK-981 abolished Ulk4 condensation and ciliary tip accumulation (Fig. 7A–C), suggesting that Ulk4-EE condensation and constitutive ciliary tip localization might depend on its SUMOylation. To determine whether Ulk4-EE condensation and constitutive ciliary tip localization depend on its SUMOylation at K1030, we examined Ulk4-EE-KR–Fg in which K1030 is mutated to an arginine residue. In contrast to Ulk4-EE–Fg, Ulk4-EE-KR–Fg neither formed condensates in HEK293T cells nor accumulated at ciliary tip when expressed in NIH3T3 cells (Fig. 7D–F).

Consistent with its constitutive ciliary tip accumulation and its ability to recruit Stk36, Gli2 and Sufu to the ciliary tip (Fig. 1A,C; Fig. 2C–F), Ulk4-EE also exhibited constitutive pathway activity, as indicated by Gli2 phosphorylation and transcriptional activation of *Ptch1* and *Gli1* in the absence of Shh (Fig. 7G–L). Blocking the SUMOylation of Ulk4-EE with either TAK-981 treatment or through the K1030R mutation inhibited its constitutive pathway activity (Fig. 7G–L). Knockdown of Stk36 attenuated Shh-independent activation of *Ptch1* and *Gli1* induced by Ulk4-EE (Fig. S4A,B), suggesting that the constitutive pathway activity of Ulk4-EE depends on Stk36. Indeed, Ulk4-EE could activate Stk36 as indicated by the phosphorylation of the activation loop in the Stk36 kinase domain (p158T/pS1590; Fig. S4C) even though Stk36 is normally activated by Shh independently of Ulk4 (Zhou et al., 2023). Taken together, these results suggest that phosphorylation-mediated SUMOylation and subsequent SUMO–SIM interaction drives Ulk4 condensation that recruits Stk36 and Gli2 at ciliary tip, which is essential for the role of Ulk4 in mediating Shh signal transduction.

## DISCUSSION

How Hh converts the latent Gli transcription factor into its activator form at ciliary tip has remained an enigma in the field. Our previous study revealed that Stk36 and Ulk4 act in parallel

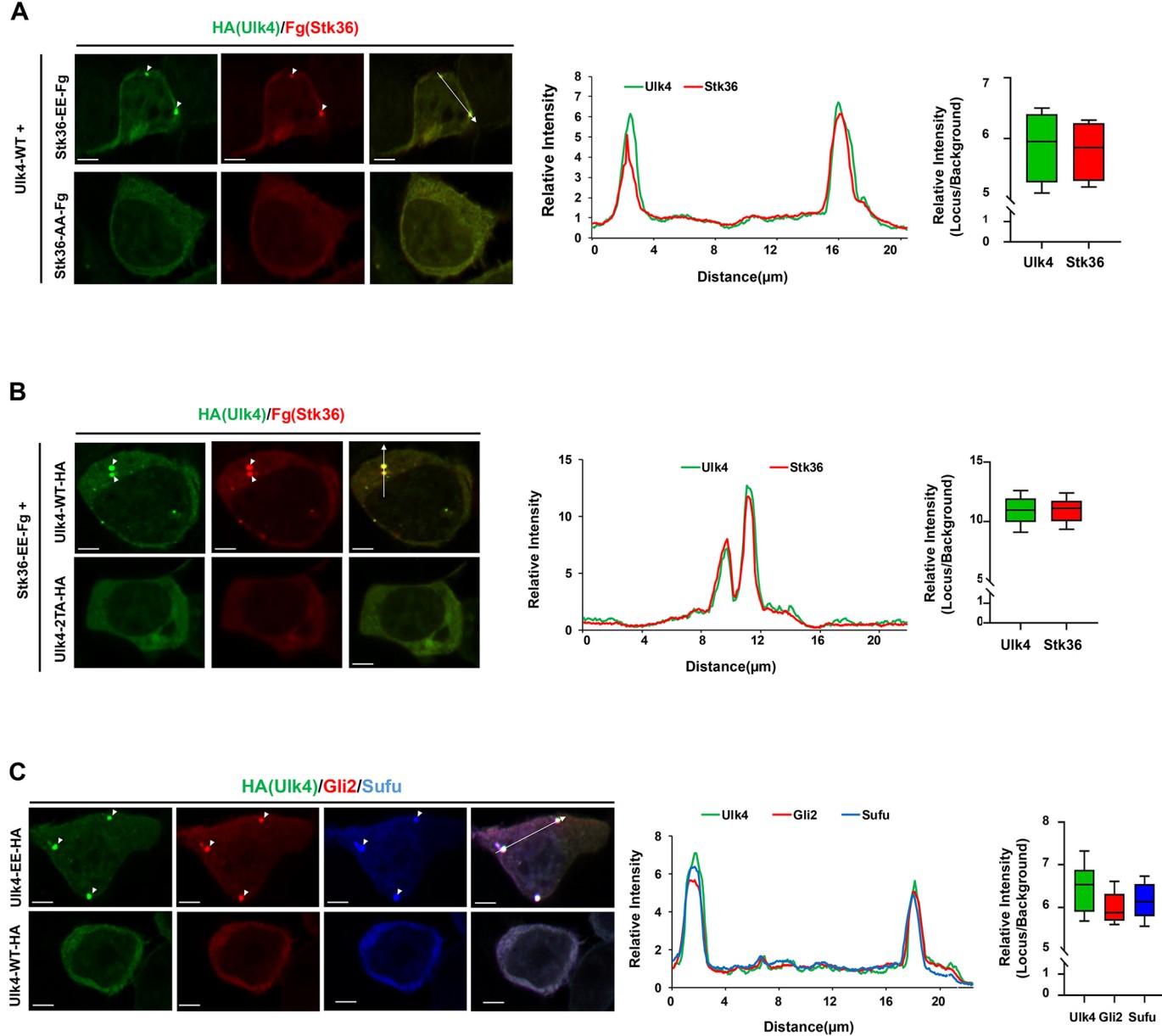

**Fig. 3. Overexpression of phosphorylated Ulk4 drives its condensation outside primary cilia.** (A,B) Representative images of immunostaining in HEK293T cells co-transfected with the indicated Ulk4–HA and Fg–Stk36 expression constructs. Arrowheads indicate Ulk4 (green in A and B) and Stk36 (red in A and B) puncta. (C) Representative images of immunostaining in HEK293T cells transfected with Ulk4-EE–HA or Ulk4-AA–HA expression construct. Arrowheads indicate Ulk4–HA (green in C), Gli2 (red in C) or Sufu (blue in C) puncta. Relative intensities and quantification of colocalization of the indicated proteins along the line shown are shown on the right. The box represents the 25–75th percentiles, and the median is indicated. The whiskers show the complete range. The background intensity is set to 1 (*n*=15 puncta).

with Ulk3 to promote Gli2 phosphorylation and activation in a manner dependent on ciliary tip localization of Stk36 and Ulk4, and that Shh-induced ciliary tip accumulation of this kinase–pseudokinase pair depends on phosphorylation of Ulk4 by Stk36 (Zhou et al., 2023). Here, we demonstrate that phosphorylation of Ulk4 induces its SUMOylation and that subsequent SUMO–SIM-mediated protein–protein interaction promotes ciliary tip condensation of Ulk4 that also recruits Stk36 and Gli2–Sufu, leading to Gli2 phosphorylation and activation (Fig. 8). We further showed that Shh-induced SUMOylation of Ulk4 depended on Ulk4 phosphorylation at T1021/T1023 and that a phospho-mimetic Ulk4 variant (Ulk4-EE) formed condensates that recruited Stk36

and Gli2 in the absence of Shh, leading to constitutive pathway activation.

Biomolecular condensates have been implicated in many cellular processes, where they function to facilitate biochemical reactions by simultaneously concentrating substrates and enzymes in the same subcellular compartments (Banani et al., 2017; Shin and Brangwynne, 2017; Su et al., 2021). The formation of biomolecular condensates through phase separation is often constitutive or regulated in response to stress signals. Whether phase separation can be induced by extracellular signals, such as morphogens, has been largely underexplored. Here, we showed that Shh induces the formation of Ulk4 condensates that recruit Stk36 and Gli2 at ciliary tip to facilitate Gli2 phosphorylation and activation, providing a

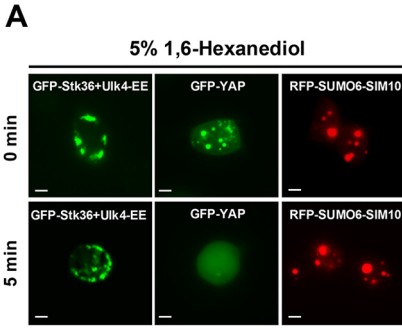

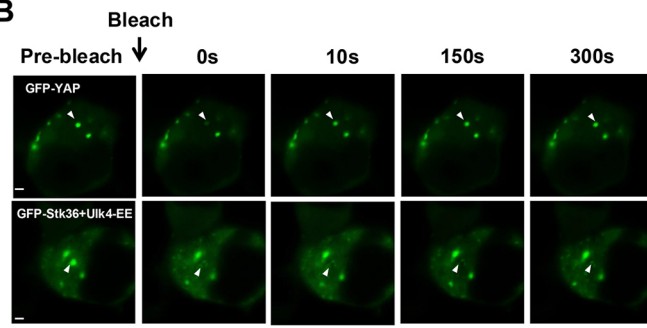

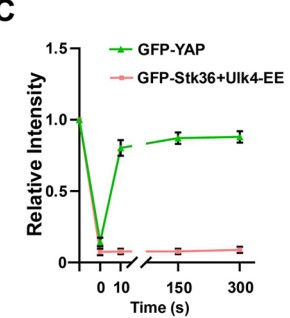

**Fig. 4. STK36 and Ulk4 form gel-like or solid-state condensates.**
(A) Representative images (from three experimental repeats) of HEK293T cells expressing the indicated constructs before and after treatment with 5% 1,6-hexanediol for 5 min. Scale bars: 6 µm. (B,C) Representative images (B) and quantification (C) of FRAP assay with mEGFP–YAP or mEGFP–Stk36+Ulk4-EE expressed in HEK293T cells. Arrowheads indicate the area of photobleaching. Data are shown as mean±s.d., $n$=5 cells. Scale bars: 2 µm.

mechanistic insight into how Shh activates Gli proteins at ciliary tip. The formation of Stk36–Ulk4–Gli2 condensates at the ciliary tip in response to Shh stimulation is likely due to local activation of Stk36 kinase by activated Smo accumulated at primary cilia and a relatively high concentration of Stk36, Ulk4, Gli2 and Sufu at ciliary tips. Indeed, overexpression of Ulk4-EE or Stk36-EE with Ulk4 in HEK293T cells led to the formation of Stk36 and Ulk4 condensates in the cytoplasm that could also recruit endogenous Gli2 and Sufu.

In most cases, the formation of membraneless biomolecular condensates is driven by LLPS, mediated through intrinsically disordered regions (IDRs) or low-complexity domains (LCDs) (Banani et al., 2017; Shin and Brangwynne, 2017). Here, we showed that phosphorylation-regulated SUMOylation of Ulk4 and subsequent SUMO–SIM interaction is responsible for Shh-induced Stk36, Ulk4 and Gli2 condensation at ciliary tip, which provides a mechanism that links kinase-mediated signaling and biomolecular

condensation. Because phosphorylation-induced SUMOylation motifs are present in many regulatory proteins (Yang and Gregoire, 2006), and SUMOylation has been implicated as a mechanism for condensate formation (Keiten-Schmitz et al., 2021), we speculate that phase separation driven by phosphorylation-dependent SUMOylation could be generally utilized in other settings and might represent a more general mechanism for kinase-mediated signaling. Indeed, our recent study has revealed that Fu and Ulk3 also form condensates in a manner that requires phosphorylation-dependent SUMOylation and SUMO–SIM interaction (Han et al., 2025). Hence, phosphorylation-dependent and SUMO–SIM-mediated kinase condensation represents a unified mechanism for Hh signal transduction from *Drosophila* to mammals, despite their differential requirement for primary cilia.

Unlike liquid-like condensates whose formation is mediated by IDRs or LCDs and can be disrupted by 1,6-hexanediol (McKnight, 2024; Sabari et al., 2018), Ulk4–Stk36–Gli2 condensates are resistant to 1,6-hexanediol treatment (Fig. 4A; Fig. S2), and thus might resemble gel-like or solid-state condensates (Putnam et al., 2019; Woodruff et al., 2017). Interestingly, SUMO–SIM-mediated condensates of Fu and Ulk3 or a synthetic construct (RFP–SUMO6-SIM10) also exhibit a gel-like/solid-state property (Fig. 4A; Fig. S2) (Han et al., 2025). It would be interesting to determine whether other condensates whose formation is mediated by SUMO–SIM interaction, such as PLM bodies, also exhibit a gel-like or solid-state property. In the case of Fu and Ulk3, SUMO–SIM-mediated condensation drives their conformational change to allosterically activate Ci/Gli2 (Han et al., 2025). It would be interesting to determine whether Stk36–Ulk4 condensation also allosterically activates Gli2 in a future study.

Although Shh-induced SUMOylation of Ulk4 and SUMO–SIM-mediated interactions are essential for the formation of Ulk4–Stk36–Gli2–Sufu condensates, other protein–protein interaction events could also be important for the ciliary tip accumulation of these intracellular Hh signaling components. For example, the N-terminal region of Ulk4 can mediate Ulk4 self-association independent of SUMO–SIM interaction (Fig. 5J), which might contribute to Ulk4 condensation. A recent study revealed that Kif7–Gli2 interaction is essential for their ciliary tip accumulation (Haque et al., 2022). In the future, it would be interesting to explore the functional relationship between Kif7 and the Ulk family kinases in the regulation of ciliary accumulation of Gli proteins.

## MATERIALS AND METHODS
### DNA constructs
Human Ulk4 cDNA was used to generate all the Ulk4 constructs used for this study. Myc–Gli2 was described previously (Han et al., 2019). Myc–Stk36, Fg-Stk36-WT, Fg-Stk36-EE, Ulk4-WT–HA, Ulk4-AA–HA, GST–Ulk4C-WT, GST–Ulk4C-2TA were generated previously (Zhou et al., 2023). N-terminally tagged mEGFP–Stk36, C-terminally tagged 3×HA-tagged human Ulk4-EE, Ulk4-KR, Ulk4-ΔSIM and C-terminally 3×Fg-tagged Ulk4-EE, Ulk4-EE-KR, Ulk4-C272-EE were subcloned into the *FUGW* vector (Addgene #14883) between EcoRI and BamHI sites. All GST fusion proteins were subcloned into the *pGEX 4T-1* vector (Cytiva) between EcoRI and XhoI sites. Ulk4C-KR, Ulk4-C272-EE, Ulk4-KR, Ulk4-ΔSIM, Ulk4-EE and Ulk4-EE-KR plasmids were constructed by PCR-mediated site-directed mutagenesis.

### Cell culture, transfection and Shh treatment
NIH3T3 cells (ATCC, CRL-1658) were cultured in Dulbecco's modified Eagle's medium (DMEM, Sigma) with 10% bovine calf serum (BCS, Gibco) and 1% penicillin-streptomycin (Sigma). A GenJet Plus *in vitro* DNA

Journal of Cell Science

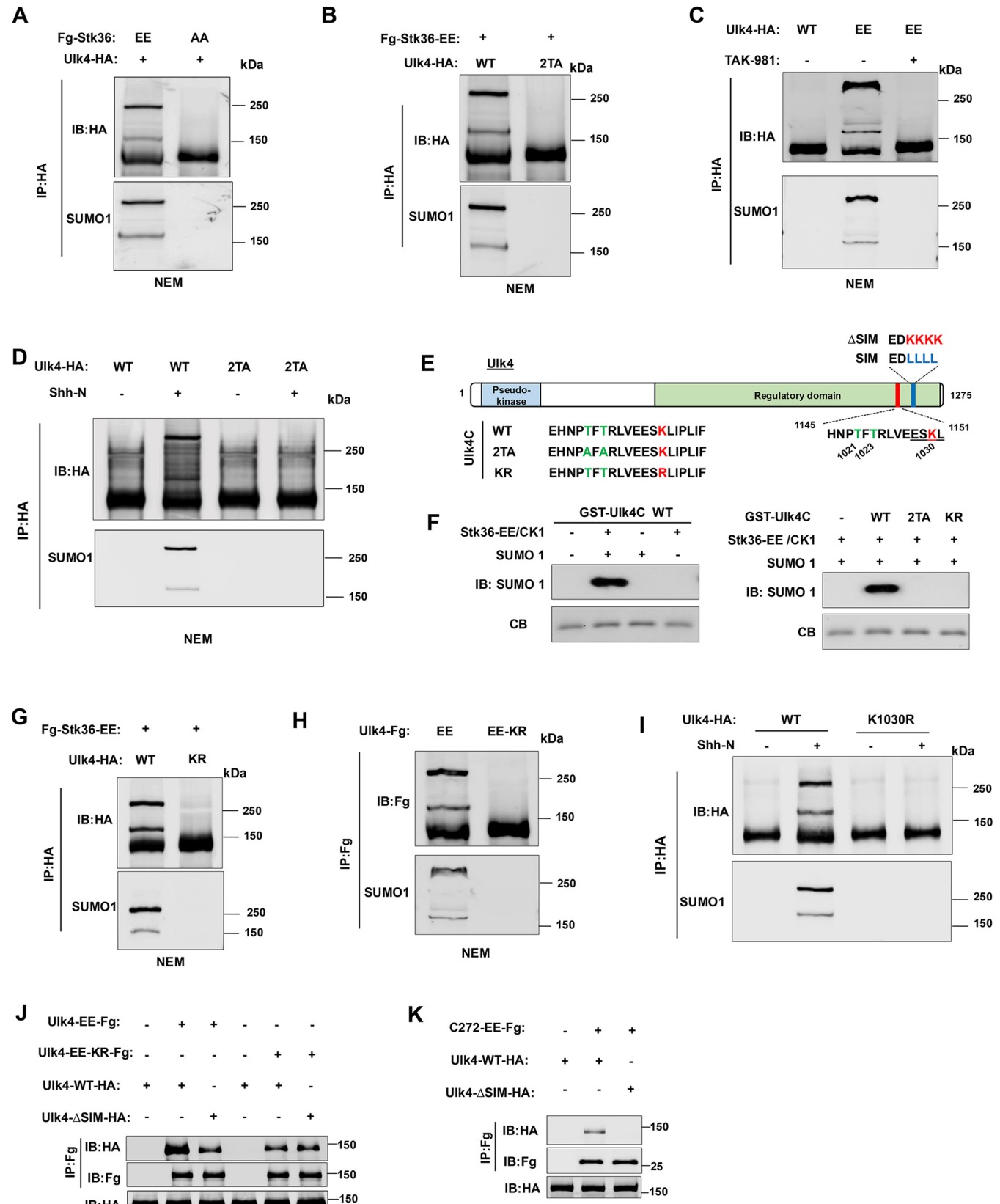

**Fig. 5.** See next page for legend.

transfection kit was used for transfection experiments according to the manufacturer's instruction (SignaGen). HEK293T cells (ATCC, CRL-11268) were cultured in DMEM containing 10% fetal bovine serum (FBS, Gibco) and

1% penicillin-streptomycin. Transfection was performed using the PolyJet *in vitro* DNA transfection kit (SignaGen) according to the manufacturer's instructions. For Shh treatment, the NIH3T3 cells were cultured at 50%

**Fig. 5. Stk36-mediated phosphorylation promotes Ulk4 SUMOylation.** (A,B) Western blot analysis of Ulk4 SUMOylation in HEK293T cells co-transfected with the indicated Fg–Stk36 and Ulk4–HA constructs. Cells were lysed in the presence of the deSUMOylation inhibitor NEM. Ulk4–HA was immunoprecipitated (IP) with anti-HA antibody, followed by western blot analysis with anti-HA and -SUMO1 antibodies. (C) Western blot analysis of Ulk4 SUMOylation in HEK293T cells transfected with a Ulk4-WT–HA or Ulk4-EE–HA construct and treated with or without TAK-981. (D) Western blot analysis of Ulk4 SUMOylation in NIH3T3 cells infected with the indicated Ulk4 constructs and treated with or without Shh-N. (E) Schematic diagram of human Ulk4 with wild-type and mutated sequences of Stk36 sites, SUMOylation site and SIM motif shown. (F) *In vitro* phosphorylation and SUMOylation of Ulk4. GST fusion proteins containing wild-type or mutant Ulk4C fragments were incubated with Fg–Stk36-EE purified from HEK293T cells and recombinant CK1, followed by *in vitro* SUMOylation by incubating with recombinant SUMO1, E1 and E2. (G,H) Western blot analysis of Ulk4 SUMOylation in HEK293T cells co-transfected with the indicated Fg–Stk36 and Ulk4–HA constructs. (I) Western blot analysis of Ulk4 SUMOylation in NIH3T3 cells infected with an Ulk4-WT–HA or Ulk4-KR–HA construct and treated with or without Shh-N. (J,K) Co-IP experiments to show Ulk4 self-association promoted by SUMO–SIM interaction. HEK293T cells were transfected with the indicated Ulk4 constructs, followed by immunoprecipitation and western blot analysis with the indicated antibodies. All images representative of three experimental repeats.

confluent densities in plate or chamber slides for 1 day and starved in serum-free medium (0.5% BCS) for 12 h to allow ciliation. The recombinant human Shh N-terminal fragment (Shh-N; 1 ng/ml; R&D Systems, #8908-SH-005) was then added to the same serum-free medium for 12 h or overnight. The cells were treated with TAK-981 (1 μM; Selleckchem, No.S8829) for 8 h before they were subjected to western blotting, RT-qPCR or immunostaining analysis.

### Immunostaining, immunoprecipitation and western blotting analysis

Immunostaining and quantification were carried out as previously described (Han et al., 2019; Zhou et al., 2023). Images were captured using a Zeiss LSM700 laser scanning microscope. Fluorescence signal intensities were measured in the respective channel by ImageJ. 20 cells were randomly picked for each experimental group to measure ciliary tip fluorescence signal intensities. Co-IP experiments were carried out as previously described (Han et al., 2019; Zhou et al., 2023). Western blotting was carried out using standard protocol as previously described (Han et al., 2019; Zhou et al., 2023). Antibodies used in this study were as follows: mouse anti-Myc (Santa Cruz Biotechnology, 9E10; 1:1000), mouse anti-Flag (Sigma, M2; 1:1000), mouse anti-HA (Santa Cruz Biotechnology, F7; 1:1000), mouse anti-acetylated tubulin (Sigma, T7451; staining 1:1000), rabbit anti-Myc (Abcam; ab9106; staining 1:2000), rabbit anti-HA (Abcam; ab9110; staining 1:500), rabbit anti-SUMO1 (4930, Cell Signaling; 1:500), rabbit anti-Sufu (Abcam, ab28083; staining 1:500), goat anti-HA (NOVUS, NB600-362; 1:400), goat anti-Gli2 (R&D system, AF3635; staining 1:1000), rabbit anti-pGli2 (pS230/232; 1:250) (Han et al., 2019; Zhou et al., 2023), rabbit anti-Ulk4 (Novus Biological, NBP1-20229, WB 1:2000), and rabbit anti-pStk36 (pT158/pS159; 1:500) (Shi et al., 2011; Zhou et al., 2023).

### SUMOylation assay

The cell-based SUMOylation assay was carried out as previously described (Han and Jiang, 2022; Ma et al., 2016). Briefly, cell pellets were lysed at 4°C for 10 min with lysis buffer containing 20 mM Tris-HCl pH 8.0, 150 mM NaCl, 5 mM EDTA, 1% IGPAL CA-630, 10% glycerol, 1 mM DTT, 20 mM NEM, 1× protease inhibitors cocktail (Roche), 1× phosSTOP phosphatase inhibitors (Roche). SDS was added into the lysates to a final concentration of 1% and the mixture was boiled for 5 min. After a 5-fold dilution with cool lysis buffer, the mixture was subjected to immunoprecipitation and western blot analysis.

*In vitro* SUMOylation experiments were performed using the SUMOylation assay kit (Abcam; ab139470) following the

manufacturer's instructions. Flag-tagged Stk36-EE was transiently expressed in HEK293T cells, purified with anti-Fg agarose (Sigma; A2220) and eluted with Flag peptide (Sigma; F4799). The GST and GST fusion proteins were purified with glutathione beads (GE; 17-0756-01) and the resulting agarose were subjective to phosphorylation by Stk36-EE and CK1 as previously described (Zhou et al., 2022). The products of *in vitro* kinase assays were used as the substrates for *in vitro* SUMOylation assays.

### 1,6-Hexanediol treatment

Cultured HEK293T cells were transfected with GFP–Stk36 and Ulk4-EE–HA, GFP–YAP or RFP–SUMO6-SIM10 (a gift from the laboratory of Michael Rosen, UTSW) DNA constructs. After 48 h, cells were treated with 5% 1,6 hexanediol for 5-30 min. Images were taken before and after treatment.

### FRAP assay

FRAP assays were carried out with a Nikon A1R confocal microscope (Ti-E) equipped with a Plan-Apo 60×/1.4 oil objective lens (Han et al., 2025). Briefly, the HEK293T cells were transfected with the mEGFP–YAP or mEGFP–Stk36 plus Ulk4-EE constructs separately. After 24 h, the cells were subjected to FRAP assays by quickly photobleaching the region of interest to ∼30% of its original intensity. The fluorescence images were immediately captured before and after photobleaching every 10 s for 300 s and analyzed using ImageJ. The results were quantified by the GraphPad Prism software.

### Lentivirus production and infection

To package the lentivirus, Ulk4 shRNA vector or *FUGW* lentiviral constructs were co-transfected with packaging plasmids (psPAX2 and pMD2.G; Addgene plasmid #12260 and Addgene plasmid #12259) into HEK293T cells using PolyJet. After 48 h to 72 h of incubation, the supernatant was transferred to a 15 ml centrifuge tube and centrifuged at 600 $g$ for 15 min. After being filtered through a 0.45 μm filter, the supernatant was collected into a new sterile tube (Beckman; 344059) and centrifuged at 20,000 $g$ for 2 h. The sediment was re-suspended, aliquoted in a small volume of 10% FBS culture medium and stored at −80°C for future use. For cell infection, lentiviruses were added to 70% confluent wild-type NIH3T3 cells together with Polybrene (Sigma, cat. no. H9268) and left overnight to get stable NIH3T3 cell lines.

### RNA interference

The mouse Ulk4 shRNA lentiviral vector was purchased from Sigma (Sigma; TRCN0000328268). Non-target shRNA plasmid DNA (Sigma; SHC016) was used as a control. The target sequence of mouse Ulk4 shRNA was: 5′-CTGCGAAGATTATCGAGAATG-3′. The sequences for Stk36 siRNA were: 5′-GGUAUACUGGCUUCAGAAA-3′ (cat. no. SASI_Mm02_00345637) and 5′-GCCUUAUGUGCUUUGCUGU-3′ (cat. no. SASI_Mm01_00167751). The sequence for the negative control was 5′-UUCUCCGAACGUGUCACG U-3′. The knockdown efficiency was validated via real-time PCR.

### Quantitative RT-qPCR

Total RNA was extracted from $10^6$ cells using the RNeasy Plus Mini Kit (Qiagen; 74134), and cDNA was synthesized with a High-Capacity cDNA Reverse Transcription Kit (Applied Biosystems; 01279158) and quantitative (q)PCR was performed using the Fast SYBR Green Master Mix (Applied Biosystems; 2608088) and a Bio-Rad CFX96 real-time PCR system (Bio-Rad; HSP9601). RT-qPCR was performed in triplicate for each of three independent biological replicates. Quantification of mRNA levels was calculated using the comparative CT method. Primers were: Gli1, 5′-GTGCACGTTTGAAGGCTGTC-3′ and 5′-GAGTGGGTCCGATTCT-GGTG-3′; Ptch1, 5′-GAAGCCACAGAAAACCCTGTC-3′ and 5′-GCCG-CAAGCCTTCTCTACG-3′; Ulk4, 5′- ATGCAGAGTGTGATTGCGTTG-3′ and 5′- GAGTGGCAGTTTCTGTGAACA-3′; Stk36, 5′-CGCATCC-TACACCGAGATATGA-3′ and 5′- AAATCCAAAGTCACAGAGC-TTGA-3′; GAPDH, 5′-GTGGTGAAGCAGGCATCTGA-3′ and 5′-GCCATGTAGGCCATGAGGTC-3′.

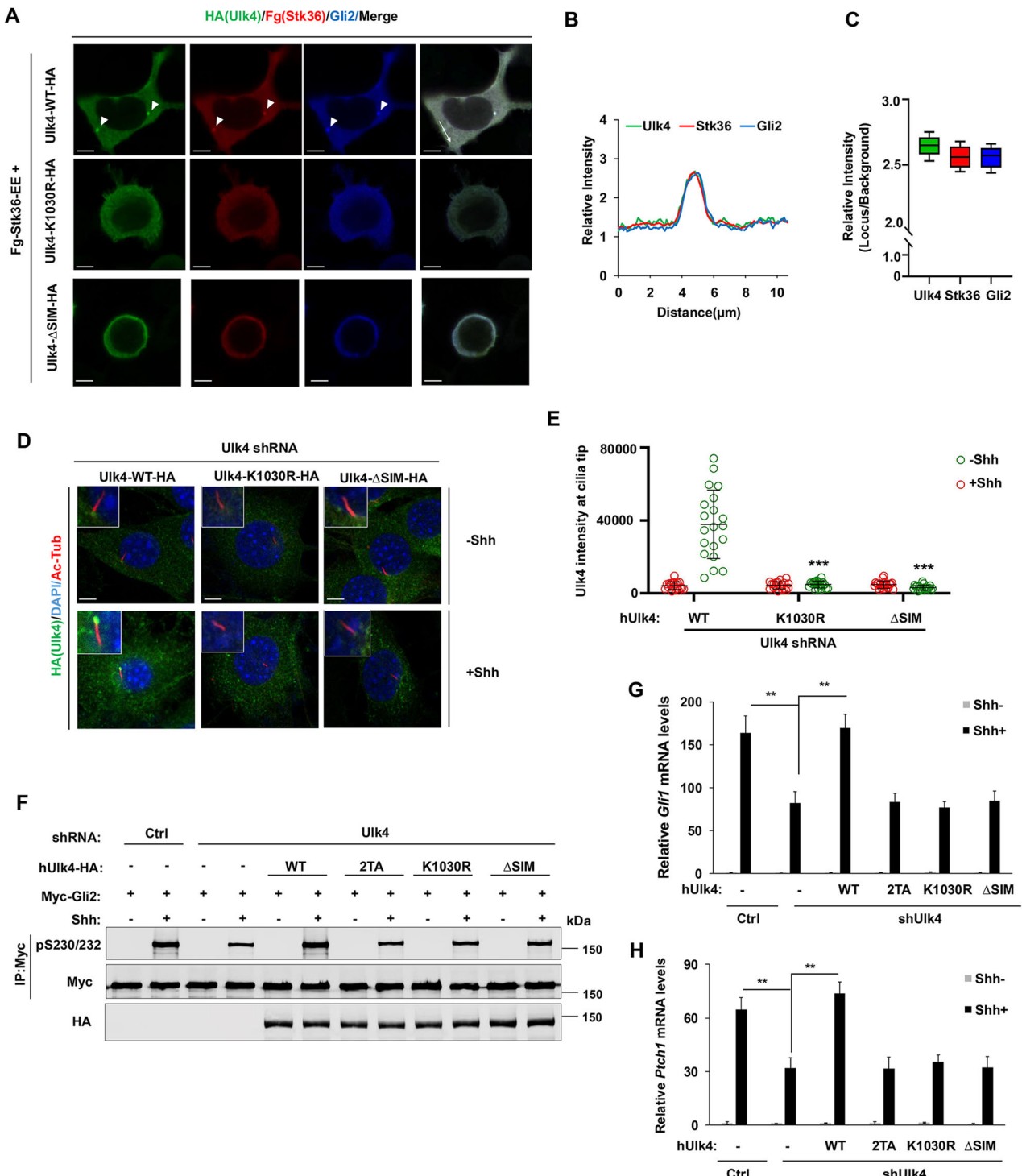

**Fig. 6. SUMOylation promotes Ulk4 condensation, ciliary tip accumulation, and pathway activation dependent on SIM.** (A) Representative images of immunostaining of HEK293T cells co-transfected with Fg–Stk36-EE and Ulk4-WT–HA, Ulk4-KR–HA or Ulk4-ΔSIM-HA constructs. Arrowheads indicate Ulk4 (green in A), Stk36 (red in A), or Gli2 (blue in A) puncta. (B,C) Relative intensities from along the line in A (B) and quantification (C) of colocalized Ulk4–HA (Green), Fg–Stk36 (Red) and Gli2 (Blue). The background intensity is set to 1 (*n*=15 puncta). The box represents the 25–75th percentiles, and the median is indicated. The whiskers show the complete range. (D,E) Representative images of immunostaining (D) and quantification (E) of ciliary tip-localized human Ulk4-WT–HA, Ulk4-KR–HA or Ulk4-ΔSIM–HA (green in D) in NIH3T3 cells infected with mouse Ulk4 shRNA and the indicated human Ulk4 lentivirus in the presence or absence of Shh-N. Primary cilia were marked by acetylated tubulin (Ac-tub) staining (red in D) and nuclei by DAPI (blue in D). The intensity of ciliary localized wild type and mutant Ulk4–HA was measured by ImageJ. 20 cells were randomly selected from each experimental group for quantification. Data are mean±s.d. \*\*\**P*<0.001 (unpaired two-tailed Student's *t*-test). Results are representatives of three independent experiments. (F) Western blot analysis of Myc–Gli2 phosphorylation on S230/S232 in NIH3T3 cells expressing the control (Ctrl) or mouse Ulk4 shRNA and the indicated human Ulk4 lentiviral constructs and treated with or without Shh-N. Image representative of three experimental repeats. (G,H) Relative *Gli1* (E) and *Ptch1* (F) mRNA levels in NIH3T3 cells expressing the control (Ctrl) or mouse Ulk4 shRNA and the indicated human Ulk4 lentiviral constructs and treated with or without Shh-N. Data are mean±s.d. (*n*=3) . \*\*P<0.01 (one-way ANOVA test with Dunnett's multiple comparisons test). Results are representatives of three independent experiments. Scale bars in A and D are 6 µm and 5 µm, respectively.

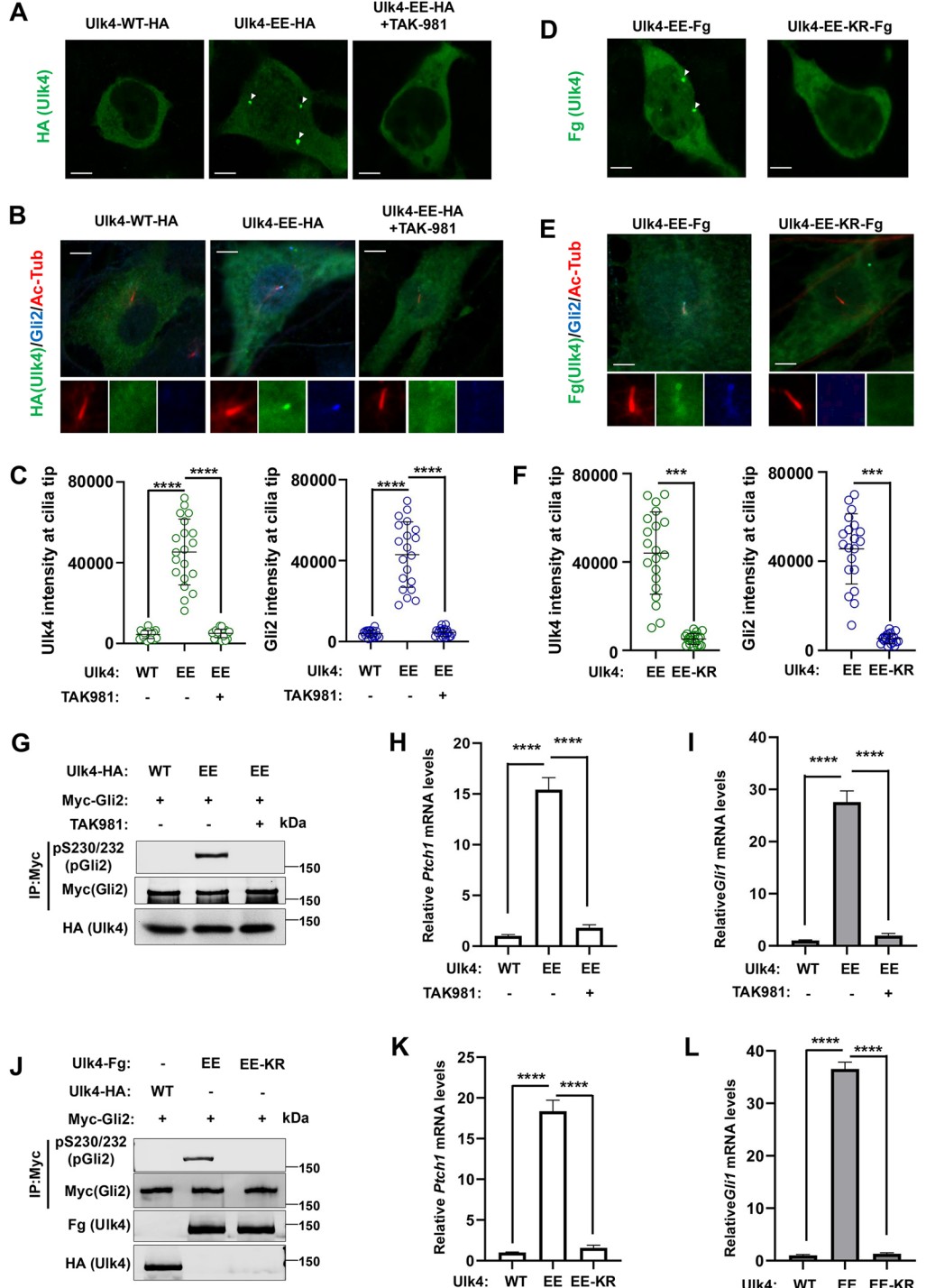

**Fig. 7. Ulk4-EE exhibits constitutive Hh pathway activity dependent on its SUMOylation.** (A) Representative images of immunostaining of HEK293T cells transfected with the indicated Ulk4 expression constructs and treated with or without the SUMOylation inhibitor TAK-981. Arrowheads indicate Ulk4 puncta. Scale bars: 6 μm. (B,C) Representative images of immunostaining (B) and quantification (C) of ciliary tip-localized Ulk4-WT–HA or Ulk4-EE–HA (green) and endogenous Gli2 (blue) in NIH3T3 cells treated with or without the SUMOylation inhibitor TAK-981. Primary cilia were marked by acetylated tubulin (Ac-tub) staining (red). Data are mean±s.d. ****$P$<0.0001 (one-way ANOVA test with Dunnett's multiple comparisons test). Results are representatives of three independent experiments. Scale bars: 5 μm. (D) Representative images of immunostaining of HEK293T cells transfected with a Ulk4-EE–Fg or Ulk4-EE-KR–Fg expression construct. Arrowheads indicate Ulk4 puncta. Scale bars: 6 μm. (E,F) Representative images of immunostaining (E) and quantification (F) of ciliary tip localized Ulk4–Fg (green) and endogenous Gli2 (blue) in NIH3T3 cells infected with Ulk4-EE–Fg or Ulk4-EE-KR–Fg lentivirus. Primary cilia were marked by acetylated tubulin (Ac-tub) staining (red). Data are mean±s.d. ***$P$<0.001 (unpaired two-tailed Student's $t$-test). Results are representatives of three independent experiments. Scale bars: 5 μm. (G,J) Western blot analysis of Myc–Gli2 phosphorylation on S230/S232 in NIH3T3 cells co-infected with Myc–Gli2 and the indicated Ulk4 lentiviral constructs and treated with or without the SUMOylation inhibitor TAK-981. Images representative of three experimental repeats. (H,I,K,L) Relative *Ptch1* (H, K) and *Gli1* (I,L) mRNA levels in NIH3T3 cells infected with the indicated Ulk4 lentiviral constructs and treated with or without the SUMOylation inhibitor TAK-981. Data are mean±s.d. ****$P$<0.0001 (one-way ANOVA test with Dunnett's multiple comparisons test). Results in are representatives of three independent experiments. The intensity of ciliary-localized wild-type and mutant Ulk4–HA was measured by ImageJ. 20 cells were randomly selected from each experimental group for quantification.

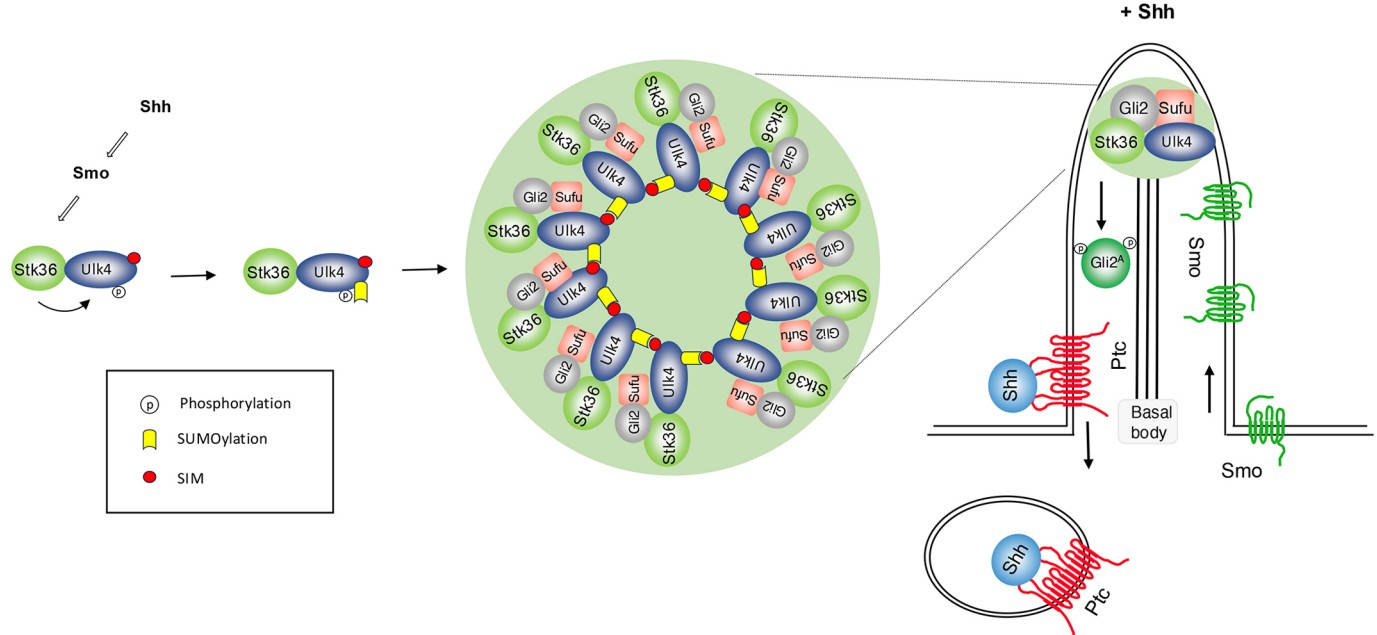

**Fig. 8. Phosphorylation-induced SUMOylation promotes kinase–substrate condensation at ciliary tip.** A working model in which Shh induces Ulk4–Stk36–Gli2–Sufu condensation at primary cilia. In response to Shh, Stk36 phosphorylates Ulk4 on T1021/T1023, which induces SUMOylation of Ulk4 on K1030. Subsequently, SUMO–SIM interaction drives the formation of Ulk4 condensates at ciliary tip, which in turn recruits Stk36, Gli2 and Sufu.

## Acknowledgements
We thank Bing Wang and Yong Suk Cho for helps during performing the experiments. J.J. is a Eugene McDermott Endowed Scholar in Biomedical Science at the University of Texas Southwestern.

## Competing interests
The authors declare no competing or financial interests.

## Author contributions
Conceptualization: J.J.; Data curation: M.Z.; Formal analysis: M.Z., Y.H.; Funding acquisition: J.J.; Investigation: M.Z., Y.H.; Methodology: M.Z., Y.H.; Supervision: J.J.; Validation: M.Z.; Visualization: M.Z., Y.H.; Writing – original draft: J.J., M.Z.; Writing – review & editing: J.J., M.Z.

## Funding
This work was supported by grants from the National Institute of General Medical Sciences (R35GM118063) and Welch Foundation (I-1603). Open Access funding provided by UT Southwestern Medical Center. Deposited in PMC for immediate release.

## Data and resource availability
All data are available in the main text or the supplementary materials.

## Special Issue
This article is part of the Special Issue 'Cilia and Flagella: from Basic Biology to Disease', guest edited by Pleasantine Mill and Lotte Pedersen. See related articles at https://journals.biologists.com/jcs/issue/138/20.

## Peer review history
The peer review history is available online at https://journals.biologists.com/jcs/lookup/doi/10.1242/jcs.263695.reviewer-comments.pdf.

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
