## [Peer Review File · Journal of Cell Science]

Phosphorylation-induced SUMOylation promotes Ulk4 condensation at the ciliary tip to transduce Hedgehog signal

Mengmeng Zhou, Yuhong Han and Jin Jiang

DOI: 10.1242/jcs.263695

Editor: Guangshuo Ou

Review timeline

Original submission:	3 November 2024
Editorial decision:	9 December 2024
First revision received:	5 March 2025
Accepted:	7 March 2025

Original submission

First decision letter

MS ID#: jcs.263695

MS TITLE: Phosphorylation-induced SUMOylation promotes Ulk4 condensation at ciliary tip to transduce Hedgehog signal

AUTHORS: Jin Jiang; Mengmeng Zhou; Yuhong Han

ARTICLE TYPE: Research Article

Dear Dr Jiang,

We have now reached a decision on the above manuscript.

To see the reviewers' reports and a copy of this decision letter, please go to:

As you will see, the reviewers gave favourable reports but raised some critical points that will require amendments to your manuscript. I hope that you will be able to carry these out because I would like to be able to accept your paper.

Reviewer 1

Advance summary and potential significance to field

In this manuscript, Zhou and colleagues show that Stk36-mediated phosphorylation of Ulk4 primes it for subsequent SUMOylation, and formation of condensates dependent on a SIM motif. They also

show that these condensates recruit Gli2 and Sufu to the ciliary tip upon Shh stimulation, and facilitate Gli2 phosphorylation and pathway activation. This manuscript provides new insights into the cilia-dependent Hh pathway activation mechanism, and is of interest to the readers of JCS.

Comments for the author

The presentation of the figures and writing of the texts are logical and easy to understand. The statistical analyses are sound. I am thus support its publication in its current form.

Reviewer 2

Advance summary and potential significance to field

This paper presents an advancement in our understanding of Hedgehog (Hh) signaling, particularly the activation of Gli2 at the primary cilium. The authors propose a novel mechanism wherein Stk36 and Ulk4 regulate Gli2 activation through phosphorylation-dependent SUMOylation of Ulk4. This post-translational modification drives the formation of biomolecular condensates, essential for Gli2 activation in response to Sonic Hedgehog (Shh). The study further demonstrates that disrupting this pathway—either by blocking SUMOylation or using SUMO-Interacting-Motif (SIM)-deficient mutants—impairs Gli2 activation. Conversely, a phospho-mimetic Ulk4 mutant promotes constitutive Hh pathway activation, underscoring the functional relevance of these modifications.

Comments for the author

The manuscript requires substantial revision to address several critical issues before publication.

Major Concerns:

1. The manuscript lacks colocalization experiments at the ciliary tip performed in HEK293T cells, which are extensively used in this study. Including these experiments would validate the relevance of HEK293T cells in modeling ciliary processes and ensure that the findings are consistent across cellular contexts.
2. The study primarily relies on overexpression systems, which can introduce artifacts. To strengthen the conclusions, several key experiments should be performed with point mutations introduced into endogenous loci of STK36 or ULK4 (e.g., using CRISPR/Cas9-mediated editing).
3. The statistical methodology is flawed. The current statistical approach pools data from three independent experiments for analysis, which is methodologically incorrect. A one-way ANOVA should be performed to account for variability between experiments and provide accurate statistical validation. Figures 1B, 1D, 2B, 2D, 5C, 5E, 5F, 6C, 6H, 6I, 6K, and 6L require reanalysis using this approach.
4. The manuscript uses inconsistent and non-standard gene and protein naming conventions, making it difficult to distinguish between species and molecular entities. Confusing naming makes the manuscript harder to follow.

Minor Concerns:

Figure 1

The rationale for deleting endogenous mouse Ulk4 using shRNA should be explicitly explained to clarify its necessity and relevance to the study. The authors should justify why not using shRNA in Figure 1C as employed in Figure 1A.

Figure 2

In Figure 2A, the authors should test whether STK36-WT colocalization at the ciliary tip depends on mouse Ulk4 by silencing endogenous mouse Ulk4. This experiment is critical to determine whether the observed ciliary localization of STK36-WT is contingent on Ulk4's presence.

In Figures 2C and 2E, the authors should clarify whether their findings establish a functional relationship between mouse Ulk4/STK36 and their human homologs.

Figure 3

Statistical analysis quantifying colocalization events in the image frames is missing and should be included.

The reliance on 1,6-Hexanediol treatment to characterize biomolecular condensates is insufficient. Additional experiments, such as fluorescence recovery after photobleaching (FRAP) or other biophysical assays, are needed to confirm the condensates' physical state and behavior.

Figure 4

In Figure 4F, if STK36-EE is a constitutively active mutant, the authors should clarify whether CK1 phosphorylates other sites to enhance its function. Additionally, the exact mutation sites comprising STK36-EE should be described.

In Figures 4J and 4K, while SIM-mediated interactions are shown to be important for condensate formation, the authors should discuss why other domains are likely required for the protein interaction, as the interaction strength observed suggests that additional structural contributions should be considered.

Figure 5 and 6

In Figure 5A, 6A and 6D, statistical analysis quantifying colocalization events is missing and should be included. This analysis is critical to support the claims made from these data.

First revision

Author response to reviewers' comments

Comments from the Reviewers:

Reviewer 1: SUMMARY OF THE ADVANCE MADE IN THIS PAPER AND ITS POTENTIAL SIGNIFICANCE TO THE FIELD

In this manuscript, Zhou and colleagues show that Stk36-mediated phosphorylation of Ulk4 primes it for subsequent SUMOylation, and formation of condensates dependent on a SIM motif. They also show that these condensates recruit Gli2 and Sufu to the ciliary tip upon Shh stimulation, and facilitate Gli2 phosphorylation and pathway activation. This manuscript provides new insights into the cilia-dependent Hh pathway activation mechanism, and is of interest to the readers of JCS.

SUGGESTIONS TO AUTHORS

The presentation of the figures and writing of the texts are logical and easy to understand. The statistical analyses are sound. I am thus support its publication in its current form.

We thank this reviewer for very positive comments.

Reviewer 2: SUMMARY OF THE ADVANCE MADE IN THIS PAPER AND ITS POTENTIAL SIGNIFICANCE TO THE FIELD

This paper presents an advancement in our understanding of Hedgehog (Hh) signaling, particularly the activation of Gli2 at the primary cilium. The authors propose a novel mechanism wherein Stk36 and Ulk4 regulate Gli2 activation through phosphorylation-dependent SUMOylation of Ulk4. This post-translational modification drives the formation of biomolecular condensates, essential for Gli2 activation in response to Sonic Hedgehog (Shh). The study further demonstrates that disrupting this pathway—either by blocking SUMOylation or using SUMO-Interacting-Motif (SIM)-deficient mutants—impairs Gli2 activation. Conversely, a phospho-mimetic Ulk4 mutant promotes constitutive Hh pathway activation, underscoring the functional relevance of these modifications.

SUGGESTIONS TO AUTHORS

The manuscript requires substantial revision to address several critical issues before publication.

Major Concerns:

1. The manuscript lacks colocalization experiments at the ciliary tip performed in HEK293T cells, which are extensively used in this study. Including these experiments would validate the relevance of HEK293T cells in modeling ciliary processes and ensure that the findings are consistent across cellular contexts.

HEK293T cells do not grow primary ciliary and are not used to study physiological response to Shh. To examine ciliary tip colocalization we used the NIH3T3 cells, which are the standard cell line in the field to study signaling events regulated by Shh. Nevertheless, due to their high transfection efficiency, HEK293T cells are commonly used for cotransfection experiments to study protein-protein interaction and protein modification and as we did in Figure 5. We also employed this system to demonstrate that over-expressed STK36/Ulk4 can form condensates in the cytoplasm (revised Fig. 3, Fig. 6A and Fig. 7A, D). For ciliary tip localization, we always stick to NIH3T3 cells and use lentiviral infection to express tagged protein at low levels (revised Fig. 1, Fig. 2, Fig. 6D, and Fig. 7B and E).

2. The study primarily relies on overexpression systems, which can introduce artifacts. To strengthen the conclusions, several key experiments should be performed with point mutations introduced into endogenous loci of STK36 or ULK4 (e.g., using CRISPR/Cas9-mediated editing).

We thank reviewer for this suggestion. However, due to the lack of a workable antibody for immunostaining to examine ciliary tip localization of endogenously expressed wild type or mutant STK36 and Ulk4 generated by CRISPR, we employed transgenic approach to express epitope-tagged wild type or mutant Stk36 and Ulk4. We used the lentiviral vector carrying a weak promoter to express tagged STK36 and Ulk4 at low levels so that they are still regulated by Shh to avoid overexpression artifact. We have examined the expression levels of exogenously expressed WT or mutant Ulk4 and found that they are comparable to that of endogenous protein. In addition, we always did experiments with mutant and wild type Stk36/Ulk4 side by side to make sure the abnormal ciliary localization of the mutant proteins is not due to overexpression artifact.

3. The statistical methodology is flawed. The current statistical approach pools data from three independent experiments for analysis, which is methodologically incorrect. A one-way ANOVA should be performed to account for variability between experiments and provide accurate

statistical validation. Figures 1B, 1D, 2B, 2D, 5C, 5E, 5F, 6C, 6H, 6I, 6K, and 6L require reanalysis using this approach.

There seems to be some misunderstanding here. Although we did multiple experiments that generate similar results, we did not pool data from multiple independent experiments but rather focused on statistical analysis of data from one experimental set. For example, in Figure 1A-B, 20 cells were randomly selected from each experimental group (Ulk4-WT without Shh or Ulk4-WT with Shh). The 20 data points (signal intensity at ciliary tip) for each group were from the same experiment and were not pooled from three independent experiments. We have made this clearer by changing the statement from “Data are mean \pm SD from three independent experiments...” to “Data are mean \pm SD. Results are representatives of three independent experiment”

We used student t test because we only compare two groups, for example, between Ulk4-WT without and with Shh or between Ulk4-EE without Shh and with Shh. We have switched to one way ANOVA when more than two groups are compared (e.g., revised Fig. 2B; Fig. 6G, H; Fig. 7C, H, I, K, L; Fig. S4A, B)

4. The manuscript uses inconsistent and non-standard gene and protein naming conventions, making it difficult to distinguish between species and molecular entities. Confusing naming makes the manuscript harder to follow.

We apologize for the name confusion. We used human ULK4 (hUlk4) cDNA to generate all the Ulk4 expression constructs because hUlk4 is resistant to the shRNA targeting the mouse Ulk4 in the knockdown experiments. For consistency and simplicity, we have now removed hUlk4 and use Ulk4 to refer to all the Ulk4 expression constructs. We added a general statement in both the main text and method to indicate that Human Ulk4 cDNA was used to generate all the Ulk4 constructs used for this study. We also indicated human Ulk4 or mouse Ulk4 in the main text and figure legends whenever it is necessary.

Minor Concerns:

Figure 1

The rationale for deleting endogenous mouse Ulk4 using shRNA should be explicitly explained to clarify its necessity and relevance to the study. The authors should justify why not using shRNA in Figure 1C as employed in Figure 1A.

We have added a sentence to explain why do the knockdown in Figure 1A.

“In this experiment, endogenous Ulk4 was depleted by shRNA targeting the mouse Ulk4 to avoid its interference with the ciliary localization of exogenously expressed Ulk4 in response to Shh”.

In Figure 1C, cells were not treated with Shh so that endogenous Ulk4 was not present at ciliary tip. The ectopic ciliary tip localization of Ulk4-EE is unlike due to the influence of endogenous Ulk4.

Figure 2

In Figure 2A, the authors should test whether STK36-WT colocalization at the ciliary tip depends on mouse Ulk4 by silencing endogenous mouse Ulk4. This experiment is critical to determine whether the observed ciliary localization of STK36-WT is contingent on Ulk4's presence.

This has been done in our previous study (Zhou et al., eLife 2023) and was mentioned in the introduction.

“Interestingly, we found that Stk36 and Ulk4 depend on each other for their ciliary localization and that ciliary tip localization of Ulk4 depends on Stk36-mediated phosphorylation of Ulk4 on T1021/T1023 (Zhou et al. 2023)”.

In Figures 2C and 2E, the authors should clarify whether their findings establish a functional relationship between mouse Ulk4/STK36 and their human homologs.

These experiments were not meant to establish that human Ulk4 can functionally replace mouse Ulk4 but rather examining whether Ulk4-EE can recruit Gli2 and Sufu to ciliary tip. Human Ulk4 and mouse Ulk4 are nearly identical in their primary sequence and should be exchangeable in term of function. As mentioned before, we used human Ulk4 cDNA to construct all Ulk4 transgenes because it is resistant to mouse Ulk4 shRNA.

Figure 3

Statistical analysis quantifying colocalization events in the image frames is missing and should be included.

We added quantifying colocalization events.

The reliance on 1,6-Hexanediol treatment to characterize biomolecular condensates is insufficient. Additional experiments, such as fluorescence recovery after photobleaching (FRAP) or other biophysical assays, are needed to confirm the condensates' physical state and behavior.

We thank reviewer for this suggestion. We have conducted FRAP experiments and included the results in revised Fig. 4.

Figure 4

In Figure 4F, if STK36-EE is a constitutively active mutant, the authors should clarify whether CK1 phosphorylates other sites to enhance its function. Additionally, the exact mutation sites comprising STK36-EE should be described.

We have included the information “CK1 was included to boost *Stk36* kinase activity in the *in vitro* phosphorylation assay because the active loop phosphorylation site T158 conforms the CK1 consensus site and previous studies revealed that phosphorylation of this conserved site in *Drosophila Stk36* homolog *Fused* (*Fu*) is essential for *Fu* kinase activation (Shi et al. 2011; Zhou and Kalderon 2011)”.

We have also indicated that the exact mutation sites comprising STK36-EE: P6 “Constitutively active *Stk36* (*Stk36-EE*: S151E/T154E; Zhou et al. 2023)”

In Figures 4J and 4K, while SIM-mediated interactions are shown to be important for condensate formation, the authors should discuss why other domains are likely required for the protein interaction, as the interaction strength observed suggests that additional structural contributions should be considered.

We thank reviewer for this suggestion and have added a sentence in the discussion:

“Although Shh-induced SUMOylation of Ulk4 and SUMO-SIM-mediated interaction are essential for the formation of Ulk4/Stk36/Gli2/Sufu condensates, other protein-protein interaction events could also be important for the ciliary tip accumulation of these intracellular Hh signaling components. For example, the N-terminal region of Ulk4 can mediate Ulk4 self-association independent of SUMO-SIM interaction (Fig. 4J), which may contribute to Ulk4 condensation”.

Figure 5 and 6

In Figure 5A, 6A and 6D, statistical analysis quantifying colocalization events is missing and should be included. This analysis is critical to support the claims made from these data.

We quantified colocalization events in Fig. 6A. Figure 7A and 7D only showed Ulk4-EE puncta.

Second decision letter

MS ID#: jcs.263695R1

MS Title: Phosphorylation-induced SUMOylation promotes Ulk4 condensation at ciliary tip to transduce Hedgehog signal

Authors: Jin Jiang; Mengmeng Zhou; Yuhong Han

Article Type: Research Article

Dear Dr Jiang,

I am happy to tell you that your manuscript has been accepted for publication in Journal of Cell Science, pending standard publication integrity checks.